# Secondary Metabolites and Their Role in Strawberry Defense

**DOI:** 10.3390/plants12183240

**Published:** 2023-09-12

**Authors:** Raghuram Badmi, Anupam Gogoi, Barbara Doyle Prestwich

**Affiliations:** 1School of Biological Earth and Environmental Sciences, University College Cork, T23 TK30 Cork, Ireland; rbadmi@ucc.ie; 2Department of Molecular Plant Biology, Norwegian Institute of Bioeconomy Research (NIBIO), 1433 Ås, Norway

**Keywords:** strawberry diseases, metabolites, terpenes, flavonoids, allergens, pathogens, beneficial microbes

## Abstract

Strawberry is a high-value commercial crop and a model for the economically important Rosaceae family. Strawberry is vulnerable to attack by many pathogens that can affect different parts of the plant, including the shoot, root, flowers, and berries. To restrict pathogen growth, strawberry produce a repertoire of secondary metabolites that have an important role in defense against diseases. Terpenes, allergen-like pathogenesis-related proteins, and flavonoids are three of the most important metabolites involved in strawberry defense. Genes involved in the biosynthesis of secondary metabolites are induced upon pathogen attack in strawberry, suggesting their transcriptional activation leads to a higher accumulation of the final compounds. The production of secondary metabolites is also influenced by the beneficial microbes associated with the plant and its environmental factors. Given the importance of the secondary metabolite pathways in strawberry defense, we provide a comprehensive overview of their literature and their role in the defense responses of strawberry. We focus on terpenoids, allergens, and flavonoids, and discuss their involvement in the strawberry microbiome in the context of defense responses. We discuss how the biosynthetic genes of these metabolites could be potential targets for gene editing through CRISPR-Cas9 techniques for strawberry crop improvement.

## 1. Introduction

Plants have evolved numerous strategies to adapt to their environments, which include various defenses against pests and pathogens. Specialized metabolites or secondary metabolites form one of the most active defense mechanisms against a wide range of pests and pathogens in several plant species. Upon attack, plants produce a repertoire of metabolites that are targeted against pests and pathogens to repel and inhibit plant invasion and reproduction. Each of these metabolites is biosynthesized by the secondary metabolite pathways in plants, resulting in a unique blend of ‘specialized arsenal’ against pathogens [1]. Through natural selection, plants have adapted to their environments for increased fitness and resilience to stresses (biotic and abiotic factors). The blend of compounds synthesized by plants is a reflection of their environments and makes up only a fraction of the total amount of compounds synthesized by the entire plant kingdom [1]. This repertoire of compounds that are biosynthesized by specialized biochemical pathways are called secondary metabolites and are referred to as ‘specialized’ metabolism [1]. The regulation of specialized metabolism is very important in determining the outcome of the crop–pathogen interactions—who wins the battle? Plants that accumulate high amounts of specialized metabolites display higher resistance against pests and pathogens than those produced in smaller amounts. Therefore, it is widely understood that exploiting this strategy is of key importance in plant protection strategies and it is important to understand strawberry (*Fragaria × ananassa*) defense in the context of its ability to synthesize specialized metabolites.

Strawberry is an agriculturally important crop and a model plant for the Rosaceae family that is vulnerable to a wide range of devastating diseases. Its widespread popularity as a favorite fruit enhances its commercial value worldwide. A recent study by Edger et al. (2019) [2] pointed out that the sub-genome derived from the diploid species *Fragaria vesca* dominates in gene number and gene expression in the octoploid strawberry *F. × ananassa,* probably due to its low-transposable element content. Furthermore, its relatively small genome size, low ploidy level, and availability of a facile in vitro regeneration and transformation system are invaluable for important investigations in plant disease research and help it serve as a model for the economically important Rosaceae crops. The genome of *F. vesca* encodes gene products that are involved in the biosynthesis of various specialized metabolites that help the plant to thrive in its natural habitats. Symbiotic microorganisms shape the plant microbiome by ‘reprogramming’ the plant’s metabolome [3], suggesting the central role played by metabolites in plant–microbe interactions. 

Plant secondary metabolites (PSM) are mainly classified into five molecular families based on their biosynthesis pathways: phenolics, terpenes, steroids, alkaloids, and flavonoids [4]. They have multiple functions such as the regulation of plant growth and development, participation in plant innate immunity, and response to environmental stresses, including defense responses to pathogens and pests [5]. The induction of defense response by PSMs is lineage-specific, tightly regulated, and possibly evolved repeatedly [6]. PSMs also act as molecular signals for establishing symbiosis between plants and microbes and, thereby, influence the composition of the microbial communities associated with the hosts [7]. This can also occur in the reverse direction, as some microbes may influence the expression of certain biosynthetic genes involved in the production of PSMs, possibly leading to the activation of the defense response against multiple pathogens [8]. There are several excellent reviews describing the role of PSMs in plant defense and their involvement in shaping the microbial communities of hosts, however, little information is available about the role of PSMs in strawberry defense and the three-way interaction of PSMs, microbes, and strawberry in enhancing the defense responses against invading pathogens. It is, therefore, important to understand strawberry (*Fragaria × ananassa*) defense mechanisms in the context of their ability to synthesize specialized metabolites in response to environmental stresses and how microbial signals act as a molecular bridge in such interactions. This article focuses on three major specialized metabolites: terpenoids, flavonoids, and allergens, and discusses their phylogenetic and regulatory aspects, including in the context of beneficial microbes. We particularly focus on the diploid strawberry *F. vesca* due to the wide availability of its genomic resources [9,10,11,12], well-understood molecular mechanisms of defense, and relatively high number of available metabolomic and transcriptomic datasets [13,14,15,16], compared to the octoploid strawberry.

## 2. Terpenoids

### Terpene Synthases

Terpenes are one of the major secondary metabolites produced by the plant kingdom that play crucial roles in plant growth, development, and resistance to biotic and abiotic stresses [17,18]. Terpenes are classified into mono-, sesqui-, di-, tri-, tetra-, and poly-terpenes based on the number of carbon atoms (C10, C15, C20, C30, C40, and multiple isoprene units, respectively) present in their structures that are formed by the condensation of individual isoprene units (C_5_H_8_). Terpenes have emerged as the major players in resistance responses against herbivores and pathogens [19,20] as well as in beneficial plant–microbe interactions [21], thus pointing to their pivotal role in mediating plant–environmental interactions. Despite their wide distribution, occurrence, and important functions, the specific roles of these compounds in major crop–pathogen interactions are poorly studied. Also, the regulation of terpene biosynthetic genes upon pathogen infection in strawberry is underrepresented in the literature. Terpene synthases and (−)-germacrene D synthases are two of the many terpene biosynthetic enzymes, the genes of which are highly upregulated upon infection by different pathogens in strawberry [14,15,22,23,24]. (−)-germacrene D synthase (GDS) catalyzes the reaction to convert farnesyl diphosphate, a precursor of sesquiterpenes, to (−)-germacrene D. Sesquiterpenes, a class of terpenes having a 15 carbon-atom (15C) backbone in their structure, are one of the important metabolites synthesized in response to plant invasion by pathogens. For example, the expression of several terpene synthase genes was induced in strawberry after inoculation with the following fungal pathogens: *Botrytis cinerea*, *Colletotrichum*
*gloeosporioides*, and *Phytophthora cactorum* [14,15,22,23,25,26,27].

As a first step towards understanding the roles of terpenes and the reg ulation of their biosynthetic genes in *F. vesca*, we analyzed the genes encoding the biosynthetic enzymes of terpenes. We constructed a phylogenetic tree of all the terpene synthases (TPS) encoded by the *F. vesca* genome by utilizing the 29 functional tomato (*Solanum lycopersicum*) terpene synthases [28]. We downloaded the *F. vesca* TPS homologs by repeated BLAST searches using the *S. lycopersicum* TPS protein sequences. Using the phylogenetic tree, we identified that the genome of *F. vesca* encodes for 43 terpene synthases (TPS), of which 25 belong to the TPS-a subfamily, 8 to the TPS-b subfamily, 6 to the TPS-c subfamily, and 2 each to the TPS-e/f and TPS-g subfamilies (Figure 1). The number of TPS encoded by *F. vesca* is more than other model plants such as *Populus trichocarpa* (32), *Arabidopsis thaliana* (33), and *Solanum lycopersicum* (29) [29], especially in the TPS-a clade. Using the annotations for the latest *F. vesca* genome [2], we found that 20 out of 25 genes in the TPS-a clade encode for (−)-germacrene D synthase (GDS) and (−)-germacrene D synthase-like (GDS-like) enzymes and 5 genes encode for (−)-alpha-pinene synthase-like (APS-like) enzymes (Figure 1, Table 1). More information about the annotations of various TPS in different clades is provided in Table 1. In the TPS-b clade, five genes encoding tricyclene synthase EBOS, chloroplastic-like genes are grouped together in one sub-clade which is separated from and one alpha-farnesene synthase and two probable terpene synthase 9-encoding genes. In the TPS-c clade, we found five genes encoding ent-copalyl diphosphate synthase (EDS) or EDS-like-encoding genes and one GDS-like gene. In TPS-e/f and TPS-g clades we found two genes each of *ent-kaur-16-ene synthase* and (3*S*,6*E*)-*nerolidol synthase 1-like* enzymes.

We used Plant-mPLoc [30] to predict the subcellular localizations of the *F. vesca* terpene synthases. Interestingly, all TPS proteins were predicted to be in the chloroplast, except for *FvH4_4g27810* which was predicted to be in the nucleus (Table 1), suggesting that the biosynthesis of terpenes could be carried out in plastids, as previously reported [31,32].

## 3. Pathogenesis-Related Proteins

Plants have evolved robust resistance mechanisms that sense the stress signals encountered and mount appropriate defense responses. Upon perception of a pathogen, plants activate a variety of defense mechanisms to counter pathogen invasion. This includes the biosynthesis of defense-related proteins, secondary metabolites, and cell-wall reinforcement. The activation of plant defenses after pathogen perception is mediated by well-known plant defense hormones such as salicylic acid (SA) and jasmonic acid (JA) [14,15,33]. The accumulation of SA and JA in the plant leads to increased levels of the pathogenesis-related (PR) proteins that help in resisting the pathogen spread and thereby offer protection to the plant [15,34]. *PR* genes are a part of the inducible plant defense responses that are highly responsive to various stresses such as cold, salinity, dehydration, UV light, wounding, and pathogen attack. The induction of *PR* gene expression is one of the most conserved responses in plants against stress factors [24,34]. Especially, the induction of *PR1* gene expression is widely used as a signature for the activation of plant defense via the pathogen-triggered systemic acquired resistance (SAR) pathway [35]. For instance, in strawberry, a basic form of the *PR1* gene was even upregulated in two resistant genotypes, while it was downregulated in a susceptible genotype after inoculation with *P. cactorum* [16].

Plants encode several PR proteins and are classified into 17 gene families (*PR-1* to *PR-17*) based on their enzymatic/biochemical functions and amino acid sequence similarities. Each family of PR proteins is known to have specific functions that are directed towards a particular pathogen type. For example, β-1,3-glucanase (PR-2) and chitinase (PR-3, PR-4, PR-8, and PR11) function in degrading the fungal cell wall components, defensins (PR-12), and thionins (PR-13); lipid-transfer proteins (PR-14) have anti-fungal and antibacterial activities; and protease inhibitors (PR-6), thaumatin (PR5), and PR1 function against oomycete pathogens [36]. The PR-10 protein family displays sequence homology to ribonucleases and was recently shown to possess ribonuclease activity in vitro [37]. However, the evidence for in vivo ribonuclease activity, its role in defense responses, and the biological functions of the PR-10 group remains unclear. Some PR-10 family proteins are constitutively expressed in pollen, fruits, and vegetables and may have allergic properties. When plant parts expressing allergy-causing PR-10 proteins enter the human gastrointestinal tract, they interact with IgE in the human body and trigger type-1 hypersensitivity in the allergic population.

These allergens belonging to the PR10 family of proteins are conserved in the plant kingdom. PR10 proteins function in a wide range of plant processes, ranging from plant development [38] to defense responses upon pathogen attack [14,22]. Their expression is also induced by fungal elicitors, wounding, and stress stimuli [34]. Interestingly, a few selected PR-10 genes are highly expressed in response to two different pathogens that infect two different plant parts, the shoots and the roots in *F. vesca* [14,22].

### Strawberry PR-10 Allergens

In strawberry (*Fragaria × ananassa*), 39 genes encoding allergens were identified that belong to the Bet v 1 superfamily and named accordingly as *Fra a* genes [39]. It was found that in people allergic to strawberries, the allergen Fra a 1A is recognized by IgE antibodies, thus triggering allergic reactions when strawberries are consumed [40]. The transcripts of *Fra a 1E* display decreased expression levels during fruit ripening and are predominantly expressed in the roots. The expression of *Fra 2* increases in ripe fruits whereas the transcript levels of *Fra a 3* were about three times higher in open flowers as compared to leaves [41]. Silencing the expression of *Fra a* genes decreased the levels of anthocyanins and downregulated the expression of chalcone synthase (*FaCHS*) and phenylalanine ammonia lyase (*FaPAL*) genes, indicating its important role in the biosynthesis of strawberry pigments [41]. The levels of anthocyanins and the genes *FaCHS* and *FaPAL* are also upregulated upon pathogen infection [22,33], thereby reinforcing the important role of *Fra a* genes in plant defense responses.

The genome of woodland strawberry (*F. vecsa*) encodes for 15 PR-10 allergens which are named *Fra v* genes [42]. The availability of the latest *F. vesca* genome provides the opportunity for updating the list of *PR-10* allergen genes. We identified new PR-10 allergens using the available PR-10 protein sequences by repeated BLAST searches in the *F. vesca* database (www.rosaceae.org). Table 2 lists all the *PR-10* genes, including the new members identified by BLAST searches of the known PR-10 genes from [42]. We also predicted the subcellular localization of the PR-10 proteins using the Plant-mPLoc server [30] which revealed that all of the PR-10 allergens, including the newly identified ones (Fra v 1.14 to Fra v 1.17), are localized in the cytoplasm except for Fra v 1.04, which is predicted to localize in the cytoplasm and the nucleus. The phylogenetic tree constructed using MEGA X [43] from the protein sequences of FvPR10 allergens grouped them according to their protein sequence similarity into different clades (Figure 2).

## 4. Flavonoids

Chemically and functionally, flavonoids are quite diverse [44] and can be divided into several major groups including flavan-3-ols, flavanones, flavones, flavanols, anthocyanidins, and isoflavones [45]. They are composed of structures with three phenolic rings from which several derivatives can be formed [46]. Three major flavonoid sub-groups have been described which include Flavonoids, Neoflavonoids, and Isoflavonoids. They are formed through two pathways in the plant, i.e., the phenylpropanoid pathway and the polyketide pathway [47]. They have been described as stress mitigators (biotic and abiotic stress) and biostimulants [46]. In plants, they not only play an essential role in the development of color and flavor in fruits but also are key actors in plant defense, including playing a protective role in relation to UV exposure. A detailed review of their biosynthesis and classification can be found here [46]. The key enzymes identified in the biosynthetic pathway of flavonoids include Phenyl ammonium lyase (PAL); Cinnamate 4-hydroxylate (C4H); 4-courmaroyl-CoA ligase (4CL); Chalcone synthase (CHS); Flavone Synthase (FNS); Flavanone 3-hydroxylase (F3H); Flavonol synthase (FLS); Dihydroflavonol reductase (DFR); Anthocyanidin synthase (ANS); and Isoflavonoid synthase (IFS) [46]. These enzymes can be targeted in metabolic engineering studies to produce plants that are tolerant/resistant to biotic and or abiotic stresses and with desirable nutritional profiles. The beneficial effects of flavonoids on human health have been well documented [48] and include anticancer, antibacterial, and antifungal activities, amongst others [45,47].

Flavonoids are known to play a key role as chemo-attractants in plant microbial interactions upon release [47] but more work is needed to fully understand how flavonoids influence microbial dynamics in the rhizosphere. In addition, many flavonoids have been identified that accumulate in the plant after pathogen infection, e.g., the flavanone sakuranetin accumulates in rice against *Fusarium*, *Magnaporthe,* and *Rhizoctonia* and 3—deoxyanthocyanidins and luteolin are seen to accumulate in sorghum against *Colletotrichum* (see [48] for a recent review in this area). Treutter et al. [44] made the distinction between preformed (i.e., those that are stored at strategic sites across the plant during development) flavonoids and those that are induced by stress at a particular time (could be either abiotic or biotic) and play a key role in plant growth [46]. The efficacy of biocontrol agents can also be improved when grown in the presence of selective elicitors. Flavonoids are one of the most prevalent secondary metabolites found in strawberry, with over 10,000 identified to date [49]. Gu et al. [50] reported a promotion in flavonoid biosynthesis in response to an upregulation of both BIA1 (BAHD acyltransferase often associated with the production of phenolic compounds) and ACT (vinorine synthase) in strawberry upon infection by *Rhodotorula mucilaginosa* following exposure to the elicitor chitosan. Using a transcriptome and metabolome analysis of strawberry, Duan et al. [24] examined >22,000 differentially expressed genes at different development stages post-infection with powdery mildew. Whilst the pathogen activated different metabolic pathways, several differential flavonoid metabolites were down-regulated in response to infection, with the most notable being quercetin, described as one of the most important flavonols in strawberry [51].

## 5. Strawberry Microbiome Composition and Its Role in Plant Growth and Defense

Plants are associated with a plethora of diverse and complex microbial communities that influence their growth, health, productivity, and fruit quality in a beneficial, harmful, or neutral way. These microbial communities colonize different plant compartments and display different interactomes based on the genetic diversity of crop plants and their environments [52,53].

The strawberry microbiome consists of both epiphytic and endophytic microbial taxa. They are distributed in the above-ground tissues (e.g., leaves, flowers, and fruits) and below-ground compartments (e.g., roots and their rhizosphere). Studies have shown that the bacterial and fungal diversity is higher in soil and the rhizosphere compared to the phyllosphere compartments (e.g., leaves and fruits) [54,55]. Strawberry bacterial communities are dominated by phyla Actinobacteria, Alphaproteobacteria, Gammaproteobacteria, Deltaproteobacteria, and Bacteroidia [55]. Meanwhile, the strawberry mycobiome is dominated by Sordariomycetes, Dothideomycetes, Leotiomycetes, and Agaricomycetes in both plant and soil compartments [54]. Within the bacterial community, the phylum Alphaproteobacteria (formerly Proteobacteria) predominates roughly 50% of the total bacterial communities, whereas the fungal phylum Ascomycota represents 76–98% of the total fungal communities [54,56]. Some examples of epiphytic and endophytic beneficial microbes belonging to bacterial and fungal communities in strawberry are shown in Figure 3.

### Influence in Plant Growth and Resistance to Biotic and Abiotic Stress

The beneficial microbial community increases the genomic potential of a crop plant by delivering multiple functions, such as promoting plant growth and productivity and resilience to biotic and abiotic stresses. Beneficial microbes include bacterial, archaeal, and fungal communities. They promote plant growth by improving nutrient acquisition through solubilization of phosphate, increasing nitrogen availability from organic matter, and iron chelation by siderophore activity [57,58]. They also improve plant disease resistance by inducing phytohormones, production of antibiotics and fungal cell wall-degrading enzymes, and competition for iron uptake by siderophores [57] and increase tolerance to abiotic stress such as drought and salt and insect herbivory [59,60]. Such tolerance is achieved by regulating the expression of drought and salt stress-responsive genes such as *EARLY RESPONSIVE TO DEHYDRATION 15* (*ERD15*) and 1-aminocyclopropane-1-carboxylate (ACC) deaminase and modulating the plant hormonal level of ethylene and jasmonic acid [61,62].

The microbial composition in the rhizosphere is cultivar-dependent and varies depending on the type of pathogen infection. Lazcano et al. [63] reported that strawberry cultivars resistant to *Macrophomina phaseolina* (a soil-borne pathogen causing charcoal rot root) contain high abundances of beneficial rhizobacteria in the genera *Pseudomonas* and *Arthobacter*, while in the susceptible cultivars, the genera *Sphingomonas*, *Phenylobacterium*, *Xanthomonas*, *Flavobacterium*, *Mucilaginibacter*, *Aminobacter*, *Rhizobium,* and *Isoptericola* were more abundant in the rhizosphere. On the other hand, the rhizosphere of cultivars resistant to *Verticilium dahliae* (a soil-borne pathogen causing *Verticillium* wilt) contains a significantly higher abundance of the genera *Burkholderia* and *Nocardioides*, two known fungal antagonists [63]. Strawberry is a host to many beneficial bacteria species that show antagonistic activity against *Verticillium dahliae*, *Rhizoctonia solani*, *Sclerotinia sclerotiorum*, and *P. cactorum* [64]. The majority of the antagonists belong to *P. putida*, while some are *Serratia* spp. and *P. fluorescence*. These antagonistic bacteria have proteolytic activity, with isolates that are known to have chitinolytic activity [64]. A schematic representation of microbial communities present in the strawberry rhizosphere that are influenced by different pathogens and cultivars is shown in Figure 4.

The strawberry rhizosphere microbial composition can also be influenced by above-ground pathogen infection. For example, the inoculation of strawberry with the gray mold pathogen *B. cinerea* resulted in a significant increase of 31 bacterial genera in the rhizosphere. A shift in the microbial community also occurred following the addition of biochar (a by-product of pyrolysis) amendment on strawberry roots. Some of the high abundance groups are *Granulicella*, *Mucilaginibacter*, and *Byssochlamys* [65], the latter two groups have plant-growth-promoting activity and are known as biocontrol agents [66,67]. It has been suggested that plants recruit rhizosphere microbes to enhance innate immune responses against invading pathogens [65]. For example, a fungus of the genus *Trichoderma* produces secondary metabolites such as 6-pentyl-α-pyrone and harzianic acid that increase plant yield and the number of fruits upon exogenous application on strawberry roots [68]. These biologically active metabolites (BAMs) enhance the production of enzymes such as geranylgeranyl reductase, hydroxymethylglutaryl-CoA synthase, diphosphomevalonate decarboxylase, squalene synthase, and β-amyrin synthase that are involved in the biosynthesis of sesquiterpenoids and triterpenoids [68]. Such compounds (terpenoids) are mainly produced under biotic stress and are known to induce systemic resistance against invading pathogens and pests [69,70].

## 6. Potential Targets for Crop Improvement

The important role of secondary metabolism in strawberry defense opens interesting avenues for crop improvement. For example, overexpressing the (*E*,*E*)-α-farnesene synthase gene in soybeans increased the resistance against the soybean cyst nematode (SCN) by increasing the accumulation of (E,E)-α-farnesene [71]. Overexpression of the rice terpene synthase gene *OsTPS19* provides enhanced resistance against blast fungus *Magnaporthe oryzae* [72]. In pine (*Pinus massoniana*), overexpressing α-pinene synthase (*PmTPS4*) and longifolene synthase (*PmTPS21*) increases the resistance against the pine wood nematode (PWN; *Bursaphelenchus xylophilus*) [73]. By utilizing new genomic information from strawberry sequencing experiments, it is possible to find similar targets for improving disease resistance in strawberry. Since terpene biosynthetic genes are highly induced upon attack by different pathogens, the enhanced expression of these genes might offer broad-spectrum disease resistance.

Fra a 1 proteins expressed in strawberry cause oral allergic syndrome (OAS) by binding with human IgE [39,40] and are also important for the biosynthesis of strawberry pigments [33]. Studies on the Der f 2 mite allergen show that multiple mutations within the IgE-binding area decreased the allergenicity of Der f 2 without changing the global structure of the protein [74], suggesting the possibility of reduced allergenicity without affecting its functions. Decreasing strawberry Fra a 1 proteins in their IgE binding regions by point mutations might create its hypoallergenic homologs with reduced allergenicity, thus opening up opportunities to develop improved strawberry varieties.

The increased accumulation of flavonoids is associated with an increase in resistance against *B. cinerea* as well as increased shelf-life in tomato fruits [75]. Engineering the flavonoid biosynthesis genes to enhance their accumulation may result in strawberries with improved resistance profiles and longer shelf-life.

The availability of high-quality genome sequences and new gene editing techniques such as CRISPR-Cas opens exciting research opportunities for crop improvement. Using CRISPR-Cas systems, it is now possible to precisely edit up to a single nucleotide in the entire genome. A seven base-pair deletion using CRISPR-Cas9 in the tomato *SlJAZ2* (*Jasmonate Zim Domain*) gene increased resistance to bacterial speck disease caused by *Pseudomonas syringae* pv. tomato (*Pto*) *DC3000* [76]. Mutations in the grapevine (*Vitis vinifera*) *VvMLO3* (mildew resistance Locus O) gene resulted in increased resistance against powdery mildew caused by *Erysiphe necator* [77]. Genome editing using Cas12 has a few advantages over Cas9 in terms of flexibility in designing single-guide RNAs and larger sequence deletions [78]. Recently, an improved version of Cas12 was shown to have high efficiency in genome editing in barley and Brassica species [78,79]. These technological advances in genome editing combined with growing knowledge of strawberry genes provide unprecedented opportunities to revolutionize disease resistance and produce higher quality strawberries.

## 7. Conclusions

The plant secondary metabolism is an important part of a plant’s active defense against pests and pathogens. Terpenes, allergens, and flavonoids are accumulated in the plant as a response to a challenge by pathogens. Evidence suggests that the accumulation of these secondary metabolites is accompanied by the transcriptional induction of their respective biosynthetic genes in strawberry. Furthermore, secondary metabolites are also involved in plant interactions with beneficial microbes. The genes involved in the biosynthesis of terpenes, allergens, and flavonoids have the potential to be used as targets for improving the quality of strawberry plants, including disease resistance, enhanced shelf-life, and decreased allergenicity. Using new genome editing tools, such as CRISPR-Cas9, it is now possible to edit genomes with precision up to a single base pair. Improved versions of the strawberry genome sequence help us reliably choose the correct target sequence for genome editing. More research into this area is required to utilize the available resources for strawberry crop improvement.

## Figures and Tables

**Figure 1 plants-12-03240-f001:**
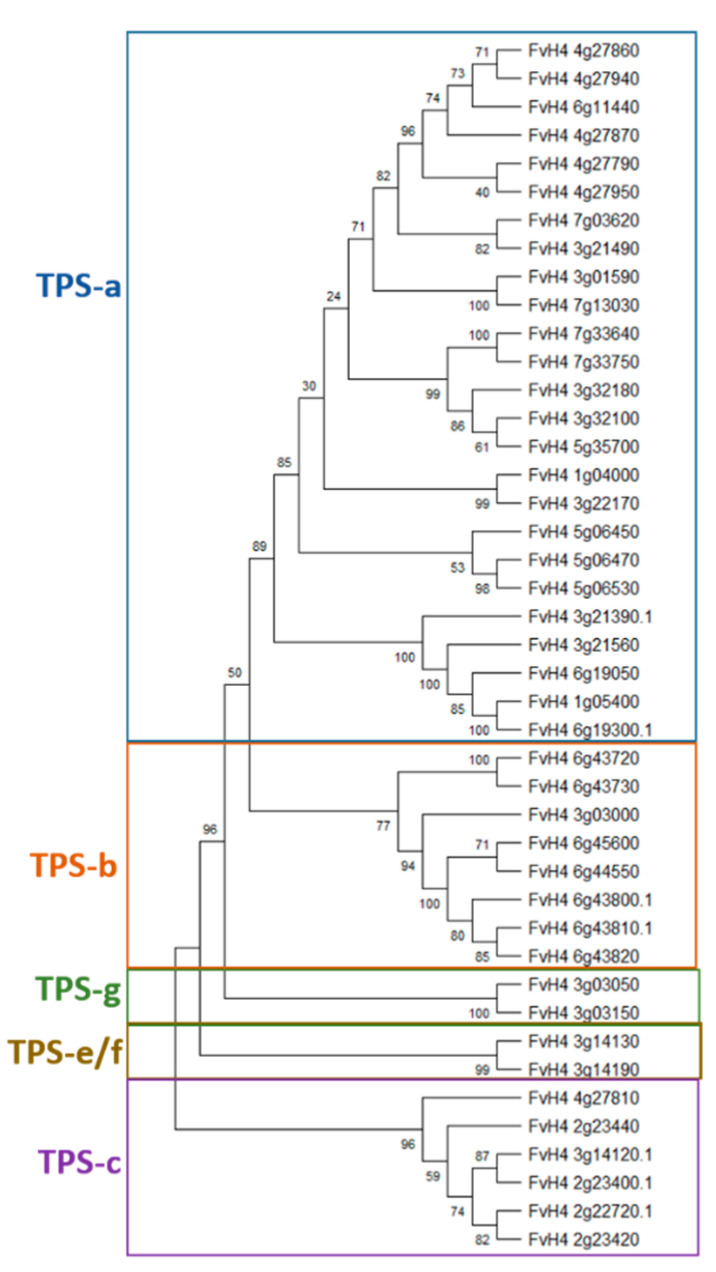
A phylogenetic tree of *Fragaria vesca* terpene synthase genes classified based on *Solanum lycopersicum* terpene synthases. The phylogenetic tree was constructed in the MEGA 10.1.7 (Molecular Evolutionary Genetics Analysis) program using the maximum likelihood method, with the protein sequence alignment as input. Bootstrap values were calculated from 1000 independent bootstrap runs.

**Figure 2 plants-12-03240-f002:**
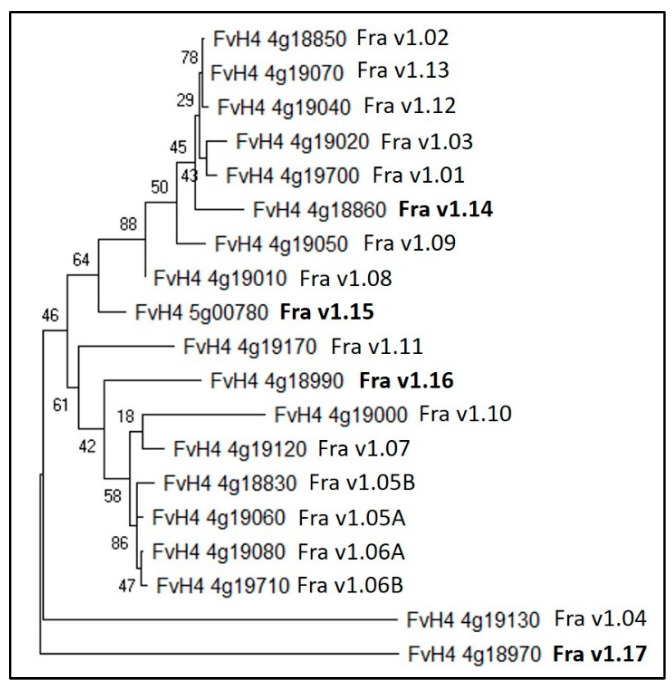
A phylogenetic tree of *Fragaria vesca* PR-10 allergen genes including the newly identified ones in this review. The phylogenetic tree was constructed in the MEGA 10.1.7 (Molecular Evolutionary Genetics Analysis) program using the maximum likelihood method, with the protein sequence alignment as input. Bootstrap values were calculated from 1000 independent bootstrap runs.

**Figure 3 plants-12-03240-f003:**
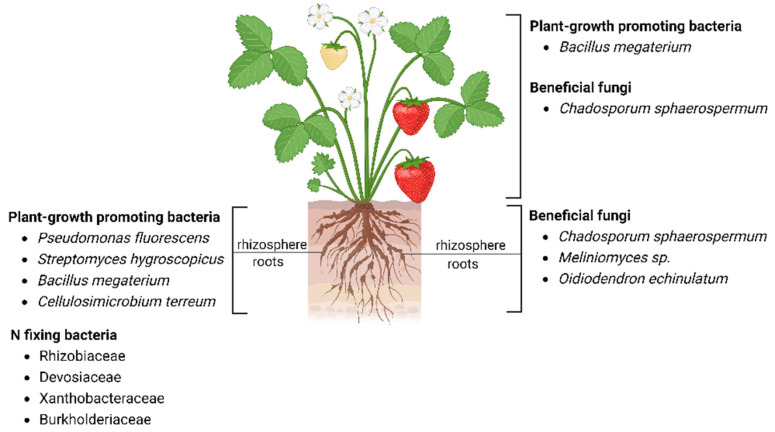
Beneficial microbial composition of the above-ground and below-ground compartments of strawberry. This figure was created with BioRender.com.

**Figure 4 plants-12-03240-f004:**
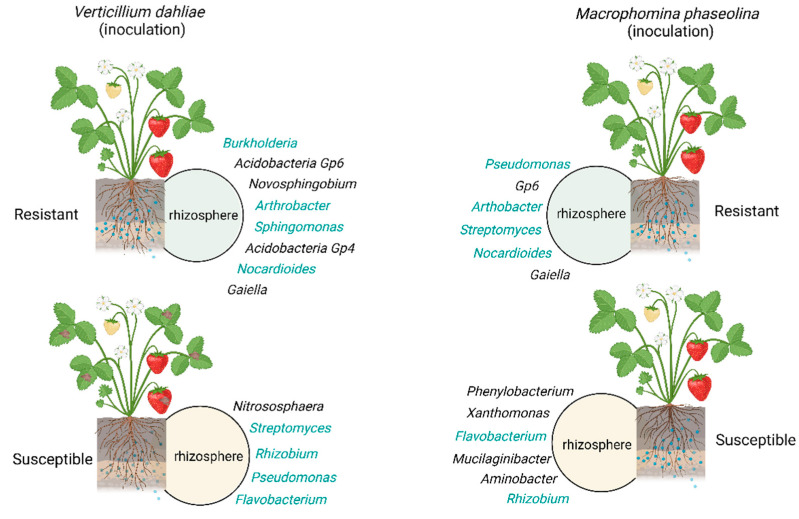
Dynamics of strawberry microbiome in the rhizosphere. The potential fungal antagonists are highlighted in dark cyan. This figure was created with BioRender.com.

**Table 1 plants-12-03240-t001:** List of terpene synthase genes and their respective IDs and annotations retrieved from the *Fragaria vesca* genome.

TPS Clade	*F. vesca* IDs	Annotations	Plant mPLOC Prediction
TPSa	FvH4_1g04000	(−)-germacrene D synthase-like	Chloroplast
	FvH4_3g01590	(−)-germacrene D synthase-like	Chloroplast
	FvH4_3g21490	(−)-germacrene D synthase-like	Chloroplast
	FvH4_3g22170	(−)-germacrene D synthase	Chloroplast
	FvH4_3g32100	(−)-germacrene D synthase-like	Chloroplast
	FvH4_3g32180	(−)-germacrene D synthase-like	Chloroplast
	FvH4_4g27790	(−)-germacrene D synthase-like	Chloroplast
	FvH4_4g27860	(−)-germacrene D synthase-like	Chloroplast
	FvH4_4g27870	(−)-germacrene D synthase-like	Chloroplast
	FvH4_4g27940	(−)-germacrene D synthase-like	Chloroplast
	FvH4_4g27950	(−)-germacrene D synthase-like	Chloroplast
	FvH4_5g06450	(−)-germacrene D synthase-like	Chloroplast
	FvH4_5g06470	(−)-germacrene D synthase-like	Chloroplast
	FvH4_5g06530	(−)-germacrene D synthase-like	Chloroplast
	FvH4_5g35700	(−)-germacrene D synthase-like	Chloroplast
	FvH4_6g11440	(−)-germacrene D synthase-like	Chloroplast
	FvH4_7g03620	(−)-germacrene D synthase-like	Chloroplast
	FvH4_7g13030	(−)-germacrene D synthase-like	Chloroplast
	FvH4_7g33640	(−)-germacrene D synthase-like	Chloroplast
	FvH4_7g33750	(−)-germacrene D synthase-like	Chloroplast
	FvH4_1g05400	(−)-alpha-pinene synthase-like	Chloroplast
	FvH4_3g21390	(−)-alpha-pinene synthase-like	Chloroplast
	FvH4_3g21560	(−)-alpha-pinene synthase-like	Chloroplast
	FvH4_6g19050	(−)-alpha-pinene synthase-like	Chloroplast
	FvH4_6g19300	(−)-alpha-pinene synthase-like	Chloroplast
TPS-b	FvH4_6g43800	tricyclene synthase EBOS and chloroplastic-like	Chloroplast
	FvH4_6g43810	tricyclene synthase EBOS and chloroplastic-like	Chloroplast
	FvH4_6g43820	tricyclene synthase EBOS and chloroplastic-like	Chloroplast
	FvH4_6g44550	tricyclene synthase EBOS and chloroplastic-like	Chloroplast
	FvH4_6g45600	tricyclene synthase EBOS and chloroplastic-like	Chloroplast
	FvH4_3g03000	alpha-farnesene synthase	Chloroplast
	FvH4_6g43720	probable terpene synthase 9	Chloroplast
	FvH4_6g43730	probable terpene synthase 9	Chloroplast
TPS-c	FvH4_2g22720	ent-copalyl diphosphate synthase and chloroplastic	Chloroplast
	FvH4_2g23400	ent-copalyl diphosphate synthase and chloroplastic-like	Chloroplast
	FvH4_2g23420	ent-copalyl diphosphate synthase and chloroplastic-like	Chloroplast
	FvH4_2g23440	ent-copalyl diphosphate synthase and chloroplastic-like	Chloroplast
	FvH4_3g14120	ent-copalyl diphosphate synthase and chloroplastic-like	Chloroplast
	FvH4_4g27810	(−)-germacrene D synthase-like	Nucleus
TPS-e/f	FvH4_3g14130	ent-kaur-16-ene synthase and chloroplastic	Chloroplast
	FvH4_3g14190	ent-kaur-16-ene synthase and chloroplastic-like	Chloroplast
TPS-g	FvH4_3g03050	(3*S*,6*E*)-nerolidol synthase 1-like	Chloroplast
	FvH4_3g03150	(3*S*,6*E*)-nerolidol synthase 1-like	Chloroplast

**Table 2 plants-12-03240-t002:** The combined list of PR-10 allergens in *Fragaria vesca* and their subcellular localization.

This Review	Hyun and Kim [42]	F_vesca_v4.0	Annotations	Plant mPLOC Prediction
	Fra v1.05B	FvH4_4g18830	Major allergen Pru ar 1-like	Cytoplasm
	Fra v1.02	FvH4_4g18850	Major allergen Pru ar 1-like	Cytoplasm
Fra v1.14		FvH4_4g18860	Major allergen Pru ar 1-like	Cytoplasm
Fra v1.17		FvH4_4g18970	Major allergen Pru ar 1-like	Cytoplasm
Fra v1.16		FvH4_4g18990	Pathogenesis-related protein PR10	Cytoplasm
	Fra v1.10	FvH4_4g19000	Major allergen Pru ar 1-like	Cytoplasm
	Fra v1.08	FvH4_4g19010	Major allergen Pru ar 1-like	Cytoplasm
	Fra v1.03	FvH4_4g19020	Major allergen Pru ar 1-like	Cytoplasm
	Fra v1.12	FvH4_4g19040	Major allergen Pru ar 1-like	Cytoplasm
	Fra v1.09	FvH4_4g19050	Major allergen Pru ar 1-like	Cytoplasm
	Fra v1.05A	FvH4_4g19060	Major allergen Pru ar 1-like	Cytoplasm
	Fra v1.13	FvH4_4g19070	Major allergen Pru ar 1-like	Cytoplasm
	Fra v1.06A	FvH4_4g19080	Major allergen Pru ar 1-like	Cytoplasm
	Fra v1.07	FvH4_4g19120	Major allergen Pru ar 1-like	Cytoplasm
	Fra v1.04	FvH4_4g19130	Major allergen Pru ar 1-like	Cytoplasm and nucleus
	Fra v1.11	FvH4_4g19170	Major allergen Pru av 1-like	Cytoplasm
	Fra v1.01	FvH4_4g19700	Major allergen Pru ar 1-like	Cytoplasm
	Fra v1.06B	FvH4_4g19710	Major allergen Pru ar 1-like	Cytoplasm
Fra v1.15		FvH4_5g00780	Major allergen Pru ar 1-like	Cytoplasm

## Data Availability

Data is contained within the article.

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
