# Peer review of "Secondary Metabolites and Their Role in Strawberry Defense"

_plants, 2023, doi:10.3390/plants12183240_

Round 1
Reviewer 1 Report (Previous Reviewer 4)
I don't think the review has always focused on strawberry research, and the references to work on other crops don't mesh well with the themes of the review.
Author Response
We thank the reviewer for their valuable comment. We would like to re-iterate the focus of this review: The literature on strawberry defences is very limited. We aimed to identify interesting areas of secondary metabolism to guide future research in strawberry. We believe that such efforts are very much needed to further the research on challenging crop plants such as strawberry.
Reviewer 2 Report (New Reviewer)
Your review: "Secondary metabolites and their role in Strawberry defense" is an interesting insight into importance of the secondary metabolite pathways in strawberry defense. Study is important because it increase overall knowledge of crop-pathogen interactions; moreover, studied plant is more than economically important species. Presented paper is written well, ideas are clear and manuscript is well stuctured. This manuscript is on a topic of enormous, it need minor revision before it ready for publication.I have some comments and suggestions:
- L12-14: in the abstract, the expression “upon pathogen attack” please avoid repetition - L26: Introduction section need recent studies, please added paragraph include more information about secondary metabolites. Increasing the bibliographical sources. - L62-64: needs further explanation and need reference. - L120: avoid revision form. - L295: "Potential Targets for Crop improvement» section should be largely improved or re-written, increasing also the bibliographical sources. - Please make figure 1, 3 and 4 more readable.
The curent review need Minor editing of English language
Author Response
We sincerely thank the reviewer for their appreciation on the topic of our manuscript. We thank the reviewer for their kind comments to improve our manuscript. We have now addressed the comments as below:
- L12-14: We have now removed the line to increase clarity
- L26: We have now included more introductory statements with recent references
- L62-64: we have now increased the description and added a reference.
- L120: We have now modified the statement as suggested and moved it to the starting part of the section for increased readability.
- L295: This section is now rewritten with additional references.
- The figures submitted in the manuscript have lost quality because of the copy paste functions. The original figures with high-quality will be submitted with the manuscript.
Reviewer 3 Report (New Reviewer)
The manuscript “Secondary metabolites and their role in Strawberry defense” provide an overview about the role of secondary metabolism in the plant defense systems. In my oppinion, the topic of the manuscript is really interesting and could contribute to improve the plant resistence agains pathogen attacks and consecuently to decrease the use of contaminnat pesticides for a more ecological agriculture and a more healthy planet.
Despite this, in my opinion there are some sections, especially those that refer to terpenes and flavonoids, that are very poorly described:
Page 2, lines 68 -73, the authors classify terpenes, but this classification is not complete since it does not include tetraterpenes or polyterpenes.
In the same section: Why don't farnesyl diphosphate synthase or geranyl diphosphate synthase appear in terpene synthases, which are key enzymes for the formation of sesqui- and diterpenes?
Pag. 5, lines 111-120. The authors suggest that all classes of terpenes are mostly synthesized in plastids, but according to the general literature, sequiterpenes and triterpenes are mostly synthesized in the cytoplasm. Could the authors detail and clarify these aspects more?
I think that this whole section would have to be rewritten for a better understanding of the readers.
Pag. 7, section 4 Flavonoides. Similar to many sequiterpenes and diterpenes, flavonoids and isoflavonoids play important defense roles in plants against pathogens. Despite its great importance, the authors only make a brief description of their chemical structures and do not mention how they are biosynthesized or the role that elicitors play in activating the key enzymes of their biosynthesis. They only mention, in the lin. 210, BAHD acyltransferase, the full name of this enzyme have to appear in the text. Vinorine synthase which leads to the formation of alkaloids???. My suggestion is that this section also has to be rewritten in more detail, consediring the important role of flavonoids in plant defense systems and the corrent knowledge about this topic.
Pag. 10, lin.292 “Such volatile compounds (terpenoids)”. Only mono-, sesqui- and some diterpenes are volatile, never tri- or tetra-terpenes. Rewrite the sentence that is too generalist.
Author Response
We sincerely thank the reviewer for their appreciation on the topic of our manuscript. We thank the reviewer for their kind comments to improve our manuscript. We have now addressed the comments as below:
- We have now included the tetra- and polyterpenes in the lines 68 – 73 to complete the classification.
- We appreciate the suggestions on farnesyl diphosphate synthase or geranyl diphosphate synthase and we realize the importance of these enzymes in sesqui- and diterpenes biosynthesis. However, in the present manuscript we aimed to focus on terpene synthases and (-)‐germacrene D synthase because the genes encoding these enzymes are highly upregulated in infected tissues in strawberry. Furthermore, these genes group under the same family in the phylogenetic tree which makes the description and analysis of these genes more focussed. We have included this reasoning in these lines for more clarity. We have also rearranged a few descriptive lines on sesquiterpenes as suggested to improve readability and flow of sentences. We thank the reviewer for their comments.
- We thank the reviewer for pointing out the lines 111-120. We have now deleted the words “all classes of” to make it more precise. We agree that not all terpenes are biosynthesized in chloroplast and this correction makes the description accurate. The description in these lines is based on our findings from the ‘Plant mPLOC’ prediction using the protein sequences of these enzymes, which is also in agreement with the cited literature.
- We thank the reviewer for their comments on section 4. We have now rewritten this section in more detail.
- More information on the chemical structure has been included with reference to a review article for further information.
- Mention is made of their biosynthesis.
- The role of elicitors in their activation is referenced.
- For the BAHD acyltransferase, a specific gene is included with the corresponding reference.
- Have reworded the sentence in relation to vinorine synthase.
- Section updated to reflect newer information.
- For line 292, we have now deleted the word “volatile” to make the sentence more accurate. Our focus in this context is to describe the accumulation of terpene compounds in defence responses of strawberry against different pathogens. Therefore, the generalist nature of this sentence is contextual and necessary to convey the appropriate message. We thank the reviewer for their valuable comments in this regard.
Round 2
Reviewer 3 Report (New Reviewer)
The manuscript has been considerably improved and in my opinion can be published in its present form.
This manuscript is a resubmission of an earlier submission. The following is a list of the peer review reports and author responses from that submission.
Round 1
Reviewer 1 Report
The work is interesting, however with some unclear issues. The title suggests that this is a review of works on some classes of chemical compounds produced by the strawberry plant and how they act in the plant's defense against pathogens, however in the work most of the topic addressed is how some compounds are important for human consumption and not how they act on the plant's defenses.
I would like it to be clear why they use Fragaria vesca as the genomic basis of the study when the hybrid that is used commercially originates from crosses between different species of Fragaria vesca, but also with the contribution of specific subspecies of Fragaria vesca, with defined geographical distribution.
The work is very general in relation to what is proposed in the objectives, which can be seen when in more than sixty references only about 20 are actually about strawberry. In this work, in fact, we did not really know the action of secondary metabolism compounds in strawberry defense processes.
Reviewer 2 Report
Dear, authors! I read over your article and found that, in addition to linguistic issues, some sections have similarities to previous studies.
Major concern:
Several of the words in this article were copied directly from other sources, so I had to stop reviewing. I won't go through every point, but before your manuscript is peer reviewed, I advise fixing any formatting issues and similarities to previous studies before resubmitting it for review in this journal.
Reviewer 3 Report
Some suggestions for the paper are as follows:
1. Don't use abbreviations in the Abstract. So, PR, should be replaced with Pathogenesis-Related;
2. L44, add the Latin name of strawberry, and L45 delete the Latin name of strawberry, L49,
3. L49, By [2] ? Should the name of the researcher be used?
4. The gene name should be in italic, such as L180 FaCHS, PaPAL.
Reviewer 4 Report
1. Line 15, Production of secondary metabolites is shaped by the microbiome of the plant and its environmental factors. The purpose of this sentence is not clear. It should be changed to a link sentence. What has been described above is that after pathogen infection, strawberry will produce the expression of secondary metabolites and induction-related genes to achieve resistance to disease. Below, beneficial microorganisms can also induce the production of secondary metabolites. Please revise the expression.
2, Line 16, Beneficial microbes also induce the production of secondary metabolites. It is not clear whether the authors mean that beneficial microbes induce the production of secondary metabolites in order to defend against pathogens. Please modify.
3, Line 118-120. The author should increase the relevant conclusion of references.
4、Line126-138, a large amount of space only quotes one reference [17], whether such a large length of text quotes is appropriate. In addition, lines 140-146, and the next paragraph, extensively describe the relationship with human health, which is not relevant to the topic of the article.